# Plate-nanolattices at the theoretical limit of stiffness and strength

Cameron Crook[1], Jens Bauer [2✉], Anna Guell Izard[2], Cristine Santos de Oliveira[3], Juliana Martins de Souza e Silva [3], Jonathan B. Berger[4,5] & Lorenzo Valdevit[1,2]

Though beam-based lattices have dominated mechanical metamaterials for the past two decades, low structural efficiency limits their performance to fractions of the Hashin-Shtrikman and Suquet upper bounds, i.e. the theoretical stiffness and strength limits of any isotropic cellular topology, respectively. While plate-based designs are predicted to reach the upper bounds, experimental verification has remained elusive due to significant manufacturing challenges. Here, we present a new class of nanolattices, constructed from closed-cell plate-architectures. Carbon plate-nanolattices are fabricated via two-photon lithography and pyrolysis and shown to reach the Hashin-Shtrikman and Suquet upper bounds, via in situ mechanical compression, nano-computed tomography and micro-Raman spectroscopy. Demonstrating specific strengths surpassing those of bulk diamond and average performance improvements up to 639% over the best beam-nanolattices, this study provides detailed experimental evidence of plate architectures as a superior mechanical metamaterial topology.

[1] Department of Materials Science and Engineering, University of California, Irvine, CA, USA. [2] Department of Mechanical and Aerospace Engineering, University of California, Irvine, CA, USA. [3] Institute of Physics, Martin-Luther-Universität Halle-Wittenberg, Halle, Germany. [4] Nama Development, LLC, Goleta, CA, USA. [5] Materials Department, University of California, Santa Barbara, CA, USA. ✉email: jens.bauer@uci.edu

The Hashin-Shtrikman (HS) and Suquet upper bounds represent the theoretical topological stiffness and strength limits of isotropic cellular solids. The effective Young's modulus ($E_{HSU}$) and yield strength ($\sigma_{y,SU}$) of a material at those theoretical limits are expressible as[1–3]:

$$\frac{E_{HSU}}{E_s} = \frac{2\bar{\rho}(5\nu - 7)}{13\bar{\rho} + 12\nu - 2\bar{\rho}\nu - 15\bar{\rho}\nu^2 + 15\nu^2 - 27} \qquad (1)$$

$$\frac{\sigma_{y,SU}}{\sigma_{y,s}} = \frac{2\bar{\rho}}{\sqrt{4 + \frac{11}{3}(1 - \bar{\rho})}} \qquad (2)$$

where $E_s$, $\sigma_{y,s}$, and $\nu$ represent the Young's modulus, yield strength and Poisson's ratio of the constituent material, respectively, and $\bar{\rho}$ is the relative density of the cellular material, i.e., the volume fraction of the solid constituent.

Over the past decade, nanolattices[1,4,5] successfully utilized ultrastrong constituent materials which approach the theoretical material strength limit; however, lattice topologies were largely limited to inefficient beam-geometries. Mechanical size effects enable nanostructures, including pyrolytic carbon octet lattices[6], nickel double gyroids[7], silica inverse opals[8,9], and gold foams[10–12] to exceed their corresponding monolithic bulk solids in strength[1]; however, their non-optimal open-cell topologies exhibit maximal performances well below the Hashin-Shtrikman and Suquet upper bounds. The discussion on mechanical efficiency of cellular materials has classically focused on the difference between stretching-dominated or bending-dominated behavior (e.g., whether lattice elements predominantly undergo uniaxial tension/compression or bending under macroscopically applied stresses, respectively), with the former being stronger and stiffer[13]. Although octet truss and isotropic truss lattices are stretching-dominated, ideal analytical models, even neglecting their nodal knockdowns, show that their stiffness and strength are limited to half the upper bounds, with the octet truss also being anisotropic[1,14,15]. In practice, stress concentrations at nodes and nodal bending further reduce their actual performance[16], to around 25% and 20% of the HS and Suquet upper bounds, respectively. Moreover, geometric manufacturing imperfections, particularly for complex and thin features, may impose further knockdowns[17]. In fact, no open-cell topology is capable of reaching the stiffness or strength[18,19] upper bounds, and while honeycombs are mechanically efficient, their response is vastly anisotropic, restricting their use to cores of sandwich structures[20].

Recently, closed-cell architectures, consisting of plates arranged corresponding to the closest packed planes of crystal structures, have computationally been predicted to reach the HS and Suquet upper bounds[18,21,22]; however, their manufacturing complexity has thus far prevented any experimental validation. The simplest such topology is the cubic+octet (CO) plate-lattice, also known as the simple cubic + face centered cubic configuration (SC-FCC), characterized by a cubic-plate unit cell embedded in an octet-plate unit cell. CO plate-lattices in theory reach the HS upper bound approaching 0% relative density and remain within 90% of the bound[18,21] at higher $\bar{\rho}$. For any given loading direction, plate-lattices have higher structural efficiency, i.e. they store strain energy more uniformly among members, and have a higher volume fraction of members favorably oriented in the loading direction, compared to a corresponding beam-lattice[18]. Moreover, the three-dimensional intersections of plates prevent the formation of kinematic mechanisms, thus ensuring the plate-lattice topology is always stretching-dominated, which may not be true in a corresponding beam topology. However, these advantages come at the cost of dramatically increased fabrication complexity. The closed-cell topologies of three-dimensional plate-lattices make most conventional fabrication routes, like assembly

techniques[16,20], impractical, and leave additive manufacturing (AM) as the only suitable approach. Nevertheless, removing raw materials enclosed within cells remains challenging. Furthermore, synthesis of a highly voided plate-lattice generally requires printing features both near the resolution and maximum build volume limits. For example, a 10% relative density CO plate-lattice requires wall features over 100 times thinner than the unit cell size, and even in fairly high-$\bar{\rho}$ plate-lattices, thin wall features are susceptible to warping. While the above challenges are a general obstacle to the application of plate-lattices, they especially complicate synthesis approaches at the nanoscale where size effects are exploitable.

In this paper, we introduce a class of nanolattices, constructed from closed-cell plate-architectures. These plate-nanolattices are the only materials to experimentally achieve the Hashin-Shtrikman and Suquet upper bounds for isotropic elastic stiffness and strength, respectively. We manufacture structures with cubic+octet design from pyrolytic carbon via two-photon-polymerization direct laser writing (TPP-DLW) and subsequent pyrolysis. Several critical fabrication challenges are overcome, including removal of excess raw material pockets, design of printing strategies to ensure homogeneous material properties for plates of different orientations and thicknesses, and management/ optimization of shrinkage during pyrolysis. Structures are characterized via micro-Raman spectroscopy, nano-computed tomography (nano-CT), and in situ mechanical compression. Approaching the theoretical limits in both the topology and constituent material properties, our plate-nanolattices demonstrate an average stiffness and strength improvement of up to 522 and 639% at a given relative density, respectively, compared to pyrolytic carbon octet-truss[6,23,24] and isotropic truss[23] nanolattices, the most advanced mechanical metamaterials reported to date. Our plate-nanolattices have the highest specific stiffness of any reported architected material and outperform all known bulk materials in compressive strength at their given densities. Reaching specific strengths of 3.75 GPa g$^{-1}$ cm$^3$, they are the only cellular materials to surpass certain diamond systems[25]. Our study clearly shows a tremendous performance improvement to be gained from plate-based topologies compared to state-of-the-art beam-based designs and provides crucial fabrication insights to enable additive manufacturing of high-performance plate-lattice materials.

## Results

**Fabrication procedure**. While manufacturing constraints have limited nanoarchitected materials to open-cell designs, like beam-lattices, we demonstrate fabricability of virtually closed-cell plate-nanolattices with dramatically improved mechanical properties over beam-nanolattices (Fig. 1 and Supplementary Note 1). TPP-DLW and subsequent pyrolysis at 900 °C were applied to create pyrolytic carbon plate-nanolattices with cubic+octet design. Relative densities of 25–60% were obtained by scaling the unit cell sizes while maintaining constant wall thicknesses of $t_c = 260$ nm and $t_o = 150$ nm of the cubic and octet plates, respectively. The ratio $t_c/t_o = \sqrt{3}$ corresponds to full isotropy at $\bar{\rho} = 40\%$[21], which is the approximate center of our examined $\bar{\rho}$ range. Given constraints on the overall build volume, all plate-nanolattices were printed at the resolution limit of the TPP-DLW process to achieve the above wall thicknesses and relative density range. This necessitated individual printing strategies for plates of different orientations to ensure octet and horizontal and vertical cubic walls had the desired thicknesses, despite the ellipsoidal voxel shape of TPP-DLW[26] (Supplementary Fig. 2). Introducing small holes with diameters of 100–160 nm at the center of the plate faces allowed infiltration with a developer and removal of

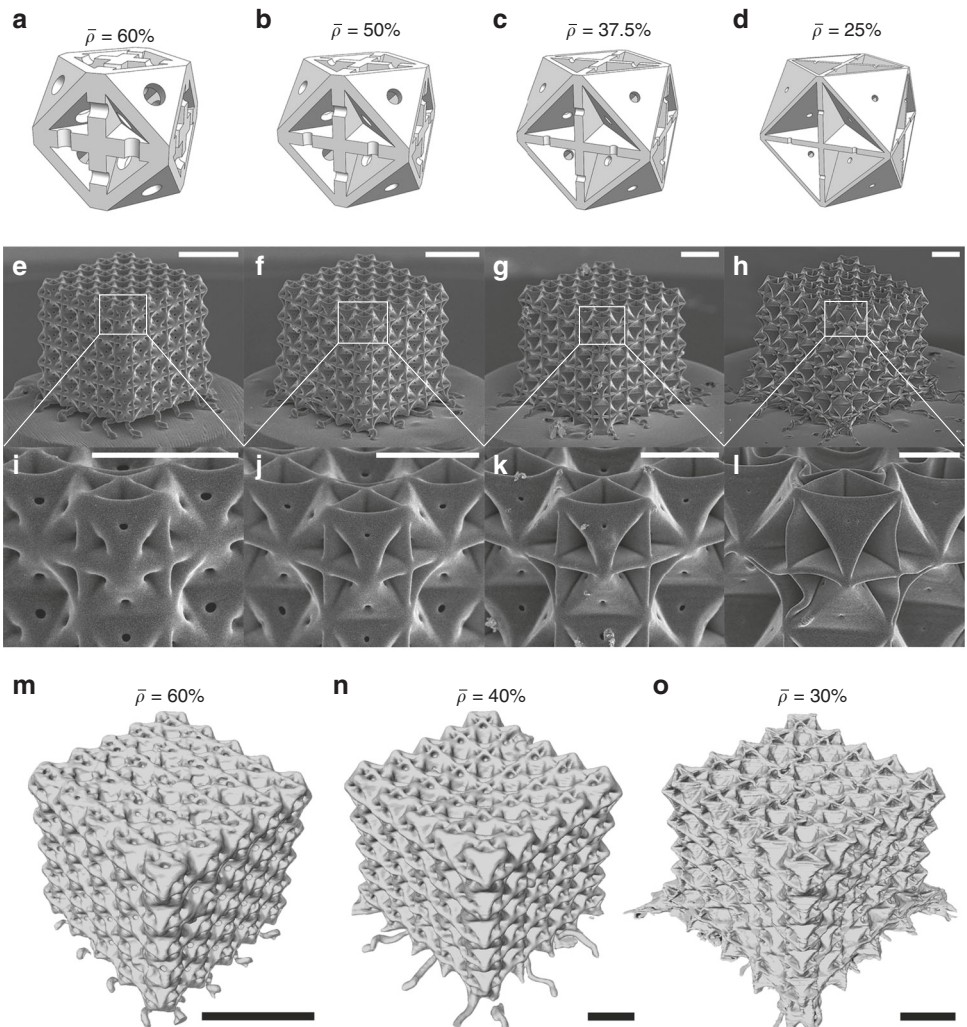

**Fig. 1 Two-photon polymerization direct laser writing (TPP-DLW) and subsequent pyrolysis creates nearly geometric defect free carbon cubic+octet plate-nanolattices.** Optimized TPP-DLW-printing strategies and nanometer-size face-holes facilitate undistorted, virtually closed-cell plate-nanolattices. Deformations occurring at the lowest relative densities ($\bar{\rho}$) are localized to the surfaces of the structures. Unit cell models (**a–d**), isometric (**e–h**) and closeup (**i–l**) SEM micrographs, and nano-CT scan reconstructions (**m–o**) for different relative densities. Scale bars are 5 μm (**e–h**, **m–o**) and 2 μm (**i–l**).

unpolymerized resin after the TPP-DLW step, despite the closed-cell topology. Nano-CT scans confirmed that the pyrolytic carbon structures were completely voided and free from residual excess material (Supplementary Movies 1–3). Final pyrolytic carbon structures with $\bar{\rho} > 37.5\%$ had near ideal, undistorted topologies. With decreasing $\bar{\rho}$, deformations occurred during pyrolysis and became more prominent at the lowest relative densities. However, nano-CT scans revealed that most deformations were limited to the specimen surfaces.

Finite element analysis showed that holes in the plate faces, introduced due to manufacturing constraints, do not significantly reduce the elastic properties of the plate-lattices nor induce any significant anisotropy (Fig. 2). As in the manufactured structures, the hole size was held constant for all modeled relative densities. Although a small performance knockdown emerged at higher $\bar{\rho}$, where the ratio of hole size-to-unit cell size was greatest, the stiffness with respect to the ideal models without holes did not fall below 96 and 93% for relative densities of 40 and 60%, respectively. All structures essentially remained isotropic despite the presence of holes, as the Zener anisotropy ratio was approximately 98% for relative densities 25–45% and around 95% at $\bar{\rho} = 60\%$. Additionally, variation of the Poisson's ratio of

the constituent material between 0.17 and 0.3 had no significant effect (Supplementary Fig. 9). In previous work, face holes were also reported not to significantly affect the yield strength of plate-lattices[18]. We numerically showed that the same applies to the buckling strength (see Supplementary Fig. 10).

**Experimental mechanical behavior.** CO plate-nanolattices with relative densities between 25 and 57.5% were in situ mechanically tested under uniaxial compression, using a nanoindentation system equipped with a flat punch tip. Figure 3 shows compressive stress-strain curves for relative densities of 57.5, 50, 37.5, and 25%, with curves recorded at all densities presented in the insets. Nonlinear behavior at small strains is attributed to slight misalignment between the structures and the indenter. A loading-unloading cycle between 4 and 10% strain was used to ensure accurate stiffness measurement, with the specimens fully seated on their supporting structures. At high relative densities, we found elastic-plastic behavior with effective stiffnesses up to 21.6 GPa, followed by brittle fracture at stresses as high as 3 GPa. Yield strains remained constant at ~5%, while failure strains, predominantly elastic, gradually decreased with decreasing relative density (Supplementary

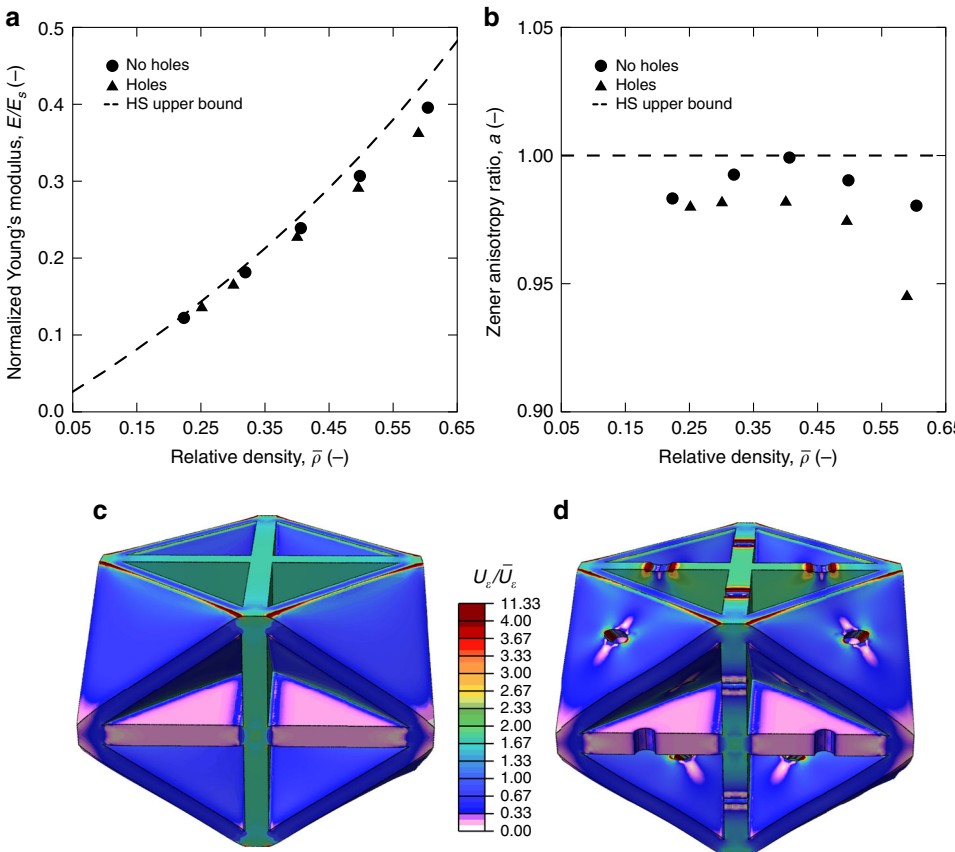

**Fig. 2 Finite element analyses of cubic+octet plate-lattice performance show that face-holes produce only a minimal knockdown effect and do not contribute strongly to anisotropy.** Effective Young's modulus (*E*) normalized by the constituent material's Young's modulus (*E*$_s$) (**a**) and Zener anisotropy ratio (*a*), (**b**), with and without holes, the dashed lines are the Hashin-Shtrikman (HS) upper bound. Simulated models of 40% relative density (**c**, **d**) show that holes do not significantly change the strain energy distribution (*U*$_\varepsilon$), here normalized by the average strain energy ($\bar{U}_\varepsilon$), while introducing local stress concentrations. Source data are provided as a Source Data file.

Fig. 7). Near $\bar{\rho} = 37.5\%$, the deformation behavior transitioned to a more progressive failure (Fig. 3c, d and Supplementary Movies 4–7) and stiffness and strength gradually decreased to approximately 5.9 GPa and 0.5 GPa respectively, at $\bar{\rho} = 25\%$.

Figure 4 shows the stiffness and yield strength results of all tested CO plate-nanolattices. The lines represent the Hashin-Shtrikman and Suquet upper bounds for different constituent material Young's moduli and yield strengths, whereby the shaded regions provide confidence intervals on the bounds given uncertainties on the material properties of nano-/microscale pyrolytic carbon. For stiffness, the lower and upper limits of the shaded region correspond to constituent Young's moduli of 41 and 62 GPa, as measured from compression of pyrolytic carbon micro-pillars printed using the same process parameters as for the vertical cubic wall (Supplementary Fig. 6) and the highest reported value for nanoscale pyrolytic carbon[27], respectively. Likewise, for the yield strength the applicable bound range is given by the minimum and maximum yield strengths measured from our micro-pillars, 2.2 GPa and 2.7 GPa, respectively, which are in good agreement with literature data on the tensile and compressive strength of nanoscale pyrolytic carbon[28,29].

Our CO plate-nanolattices experimentally reach both the Hashin-Shtrikman and Suquet upper bounds of an isotropic cellular material. Our most pristine structures, $\bar{\rho} \geq 37.5\%$, lie at the upper limit of the shaded stiffness region of Fig. 4a. For yield strength, the highest-density structures predominantly lie in the shaded region of Fig. 4b, achieving the Suquet bound; the lowest relative density structures ($\bar{\rho} < 37.5\%$) mostly lie below the bound region.

Consistent with the observed transition from linear-elastic to progressive failure and reduced mechanical properties around $\bar{\rho} = 37.5\%$, we identified two characteristic scaling relations of the measured strength and stiffness with the relative density (Supplementary Table 4). The effective stiffness of cellular materials (*E*) can be related to the relative density ($\bar{\rho}$), as $E \propto \bar{\rho}^{b}$ [30], with the scaling exponent (*b*). The effective yield strength ($\sigma_y$) can be described as $\sigma_y \propto \bar{\rho}^{c}$ with the scaling exponent (*c*). Least square fitting of our results with the above scaling relations gave $b = 1.05$ and $c = 0.85$ for $\bar{\rho} \geq 37.5\%$, and $b = 2.22$ and $c = 2.78$ for $\bar{\rho} < 37.5\%$. Near-linear scaling of both stiffness and strength with $\bar{\rho}$, for $\bar{\rho} \geq 37.5\%$, generally indicates efficient stretching-dominated behavior[21]. While scaling and deformation behavior correlation becomes less accurate above $\bar{\rho} = 30\%$[30], the observed shift to reduced scalings around $\bar{\rho} = 37.5\%$ still reveals two distinct deformation regimes with different degrees of efficiency in terms of strength and stiffness.

**Computed mechanical behavior.** Figure 5a compares the stiffnesses of pyrolytic carbon plate-nanolattices and cubic+octet finite element models with ideal and pre-deformed unit cells. For $\bar{\rho} \geq 37.5\%$, experiments and ideal finite element results excellently agree, with the computed stiffnesses coinciding with the measured stiffness-$\bar{\rho}$-scaling. The experimentally observed steepening in stiffness scaling for $\bar{\rho} < 37.5\%$ is well captured by simulations with increasing pre-deformations, with magnitudes in agreement with experimental observations. Figure 5b compares the measured yield

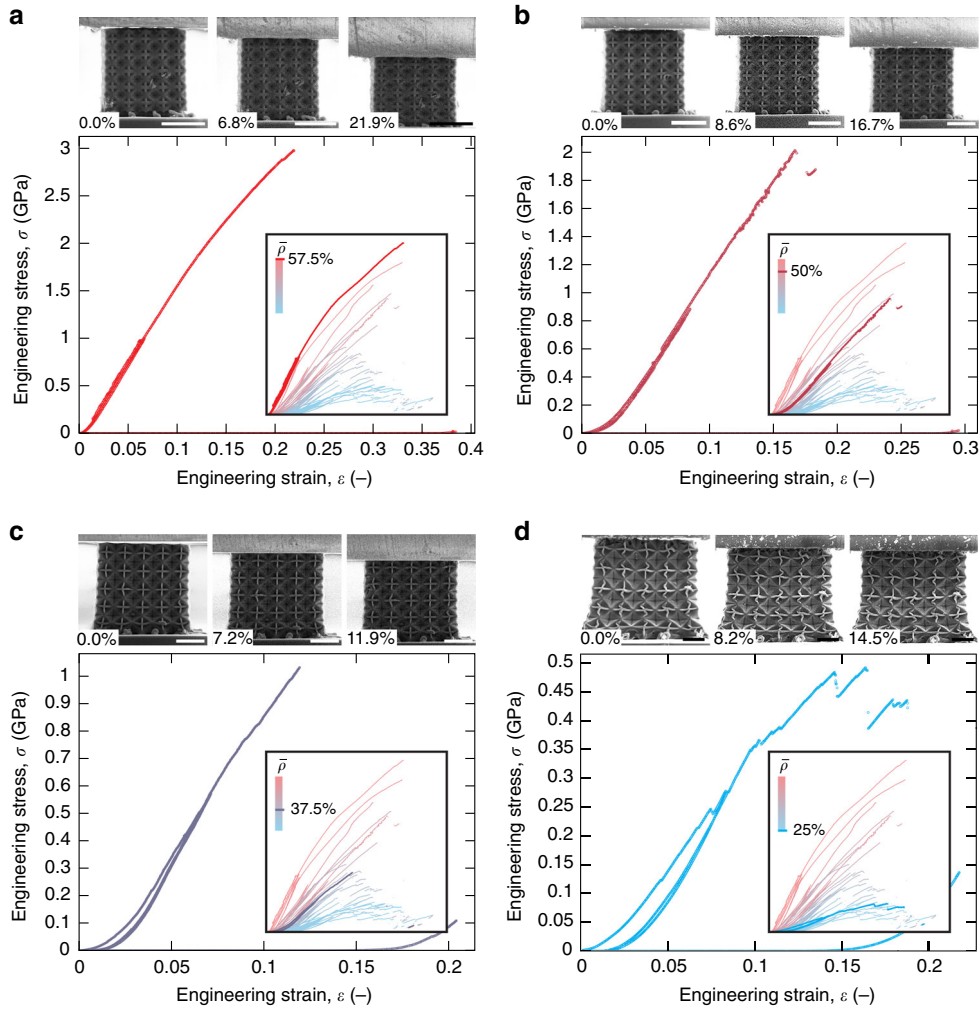

**Fig. 3 Compression experiments of pyrolytic carbon cubic+octet plate-nanolattices with different relative densities (ρ̄).** Stress–strain curves of specimens with ρ̄ from 57.5 to 25% (**a**–**d**) accompanied by front facing in situ SEM images at the indicated strains and comparison to all relative densities, show a transition from brittle fracture to progressive deformation behavior around 37.5% relative density. Scale bars are 5 μm. Source data are provided as a Source Data file.

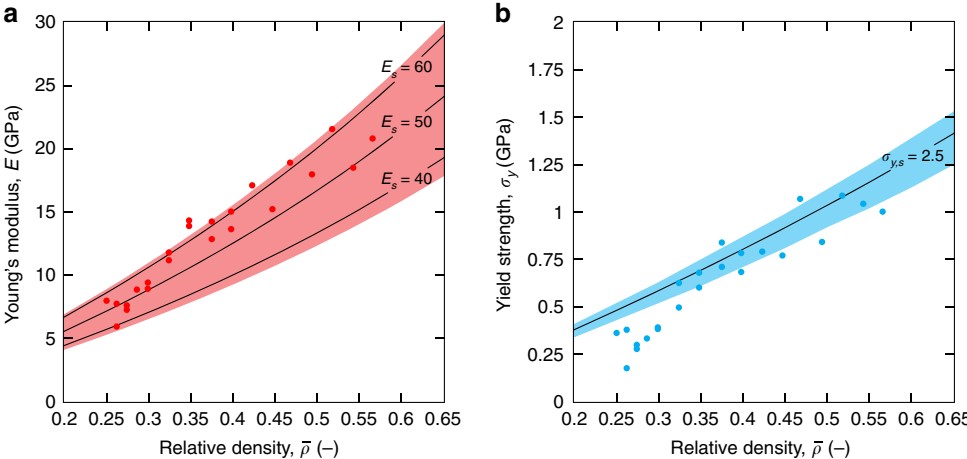

**Fig. 4 Pyrolytic carbon cubic+octet plate-nanolattices reach both the Hashin-Shtrikman and Suquet upper bounds of an isotropic cellular material for stiffness and strength, respectively.** Stiffness (**a**) and yield strength (**b**) versus relative density plots show most data points within the shaded upper bound regions, which are given by the constituent property range of pyrolytic carbon. Lines indicate upper bound functions for various constituent materials properties. Source data are provided as a Source Data file.

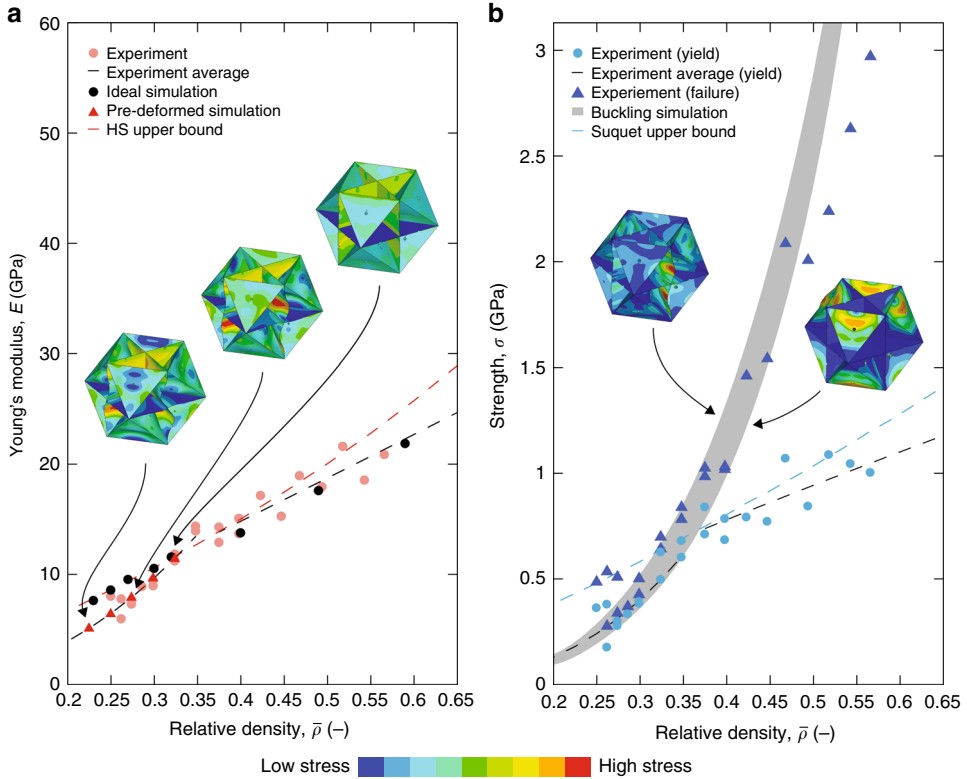

**Fig. 5 Finite element simulations confirm performance of high relative density ($\bar{\rho}$) pyrolytic carbon cubic+octet plate-nanolattices on par with theoretical predictions and show stiffness and strength reductions below $\bar{\rho} = 37.5\%$ relate to pre-deformations and premature buckling.** Computed stiffnesses of ideal and pre-deformed structures compared to the experimental data (**a**), insets show models with different degrees of pre-deformation. Computed buckling strengths of ideal models are consistent with experimental low-$\bar{\rho}$ yield strengths and suggest the compressive strength is limited by post yield-buckling above $\bar{\rho} = 37.5\%$ (**b**), the bottom and top bound of the shaded buckling strength region corresponds to the first and vertical wall eigenmodes, respectively. Source data are provided as a Source Data file.

and compressive strength to the computed elastic buckling strength of ideal models. The shaded region is bounded by the first eigenmode, occurring in the octet walls, and the vertical wall buckling mode most visible in the experiments. The computed buckling strength notably exceeds the measured yield strength in the high-$\bar{\rho}$ regime and well correlates with the measured yield strength reduction below $\bar{\rho} = 37.5\%$, indicating a transition from yielding to elastic buckling near that relative density. Across the entire $\bar{\rho}$ range the measured compressive strength trends with the computed buckling failure. Given that pyrolytic carbon only shows a modest change in stiffness between elastic and plastic deformation, the elastic buckling simulations provide a reasonable approximation for the plastic buckling strength, suggesting that high-$\bar{\rho}$ plate-nanolattices fail by plastic buckling following initial yielding. As yield and buckling strength deviate with increasing $\bar{\rho}$, prolonged post-yield deformation results in increasing failure strain with increasing $\bar{\rho}$ (Supplementary Fig. 7).

**Comparison with literature data.** With average compressive strength and stiffness improvements up to 639% and 522%, respectively, compared to the best beam-nanolattices[6,23], our plate-nanolattices are the only reported materials to lie at the theoretical compressive strength limits and the only architected material to exceed synthetic macroscale cellular materials, like ceramic foams[25], in stiffness, thus representing the strongest and stiffest existing architected materials to date. Figure 6 compares the compressive strength and stiffness data from this work with those of other architected and bulk materials[6–12,16,23,31–45]. The theoretical limits[1,6,23] are taken as regions bounded by a linear scaling of graphene (the strongest and stiffest known material,

albeit in two dimensions and at the nanoscale) and bulk diamond (the strongest and stiffest bulk material at the macroscale). With densities below 0.792 g/cm³, and compressive strengths up to 3 GPa, our plate-nanolattices achieve specific strengths up to approximately 3.75 GPa g⁻¹ cm³, which surpass all known bulk metallic, polymeric and composite materials, and all technical ceramics, including certain diamond systems[25]. With stiffnesses up to 21.6 GPa, our plate-nanolattices are the stiffest architected material reported to date. Compared to pyrolytic carbon octet truss and isotropic truss nanolattices[23] of the same relative density range, the average improvement is between 137 and 522% for stiffness and between 89 and 639% for strength, at densities of 0.792 and 0.349 g/cm³, respectively. Considering only the best performing pyrolytic carbon beam-nanolattices, we retain a 100% improvement in strength and a 33–89% improvement in stiffness. For additional comparison, normalized Ashby charts of stiffness and strength show that the cubic+octet topology of our plate-nanolattices clearly outperforms both beam and shell architectures (Supplementary Fig. 8).

## Discussion

Synthesis of nanolattices from mechanically strong and stiff ceramics or metals requires sophisticated multi-step processes that are complicated to apply to closed-cell topologies and have, so far, mostly been limited to non-optimal beam-lattice designs. High-resolution 3D additive manufacturing processes are generally limited to viscoelastic polymers, but demonstration of nanolattice performance at the Hashin-Shtrikman upper bound requires linear elastic material properties[19,21] like those of ceramics and metals (Supplementary Note 5). Ceramic and metallic

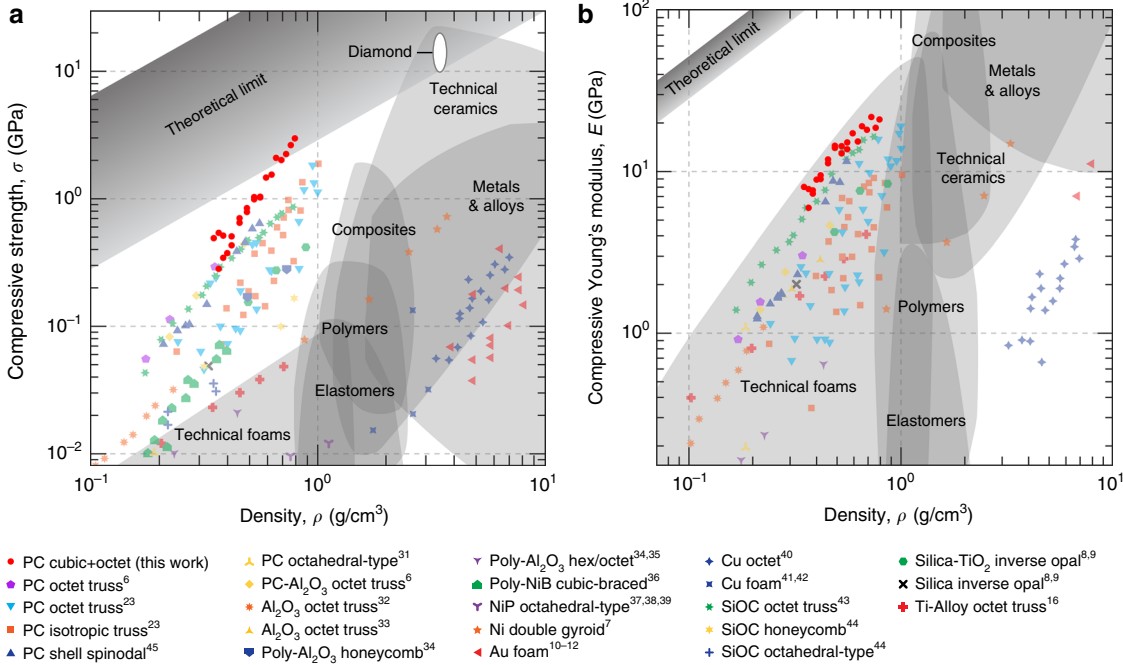

**Fig. 6 Pyrolytic carbon (PC) cubic+octet plate-nanolattices are the strongest and the stiffest existing materials for their respective densities.**
Compressive strength (**a**) and stiffness (**b**) Ashby maps. The stronger and weaker theoretical limits assume an arbitrary ideal topology with the best possible scaling of one, for two different constituent materials: graphene, the strongest known material at any scale, and diamond, the strongest known bulk material at the macroscale. With up to 639 and 522% average strength and stiffness improvements compared to the most efficient beam-nanolattices, cubic+octet plate-nanolattices are the only architected materials to surpass the bulk theoretical strength limit and to reach specific stiffnesses comparable to those of the best performing technical foams. Source data are provided as a Source Data file.

nanolattices are manufacturable by conversion[1] from polymer templates such as those printed by TPP-DLW. However, closed-cell designs impose process restrictions, complicating the adoption of atomic-layer-deposited ceramics[32,34] as well as electroless[36] and electro-plated[40] metals, even for the synthesis of composites[34–36] where templates are not removed. Pyrolysis is the only alternative, and requires structures to be fabricated such that they survive extreme linear shrinkage of up to 90%[6].

Here, we overcame several critical manufacturing challenges, leading to fabrication of highly efficient virtually closed-cell ceramic plate-topologies (Supplementary Note 1). As with most additive manufacturing techniques, TPP-DLW-printing of fully enclosed cellular geometries results in trapped excess liquid monomer and/or rupture of thin membranes during post-print development. We show that nanometer-size pores are sufficient to eliminate residual monomer from assemblies of even tens of micrometer-size cells, while retaining mechanical performance on par with fully closed cell topologies. In contrast to TPP-DLW-derived beam-nanolattices, plate-nanolattices cannot simply be printed from individual line features in one three-dimensional trajectory pattern. To address this challenge, we developed an orientation-specific layer-by-layer hatching strategy (Supplementary Fig. 2) to combine the highest surface quality with smallest possible wall thicknesses, and fully exploited size-dependent strengthening of the constituent material. Material properties of hatched TPP-DLW-derived structures are highly sensitive to printing parameters[46]; thus, carefully selected combinations of laser average power, scan speed and hatching distances were adopted herein to ensure identical constituent properties throughout our nanoarchitected material (Supplementary Fig. 3). To demonstrate property uniformity, for each plate orientation, micro-Raman spectroscopy measured nearly identical degree of conversion (DC) of pre-pyrolysis polymeric structures and nearly identical degree of graphitization (DG) for

all pyrolytic carbon structures (Supplementary Figs. 3, 5; Supplementary Tables 1, 2). While pyrolysis of TPP-DLW printed beam-lattices has been demonstrated, the much larger ratio of unit cell dimensions to feature size in plate-lattices drastically complicates this conversion by pyrolysis, e.g., the overall size of our CO plate-nanolattices is up to 5 times larger than that of same-$\bar{\rho}$ octet beam-lattices[6,23] with comparable feature sizes. Given structures with larger surface areas tend to shrink more during pyrolysis, support pillars were coarsely hatched to introduce controlled porosity to lessen the shrinkage mismatch between the large surface area cubic+octet plate-nanolattices and the monolithic support pillars.

Despite the variability in the properties of pyrolytic carbon at small scales, our cubic+octet plate-nanolattices clearly reside within the bound bands of Fig. 4. Size effects strongly increase both the strength and stiffness of pyrolytic carbon, with decreasing characteristic length[27,28,47]. In the range of 1 μm and 5 nm, Young's moduli around 40 GPa[28] and 62 GPa[27] are reported, respectively; the former agrees well with our 6 μm-diameter micro-pillar compression results (Supplementary Fig. 6). Consequently, the Young's moduli of the constituent material in our plate-nanolattices, with wall thicknesses of 150–260 nm, are expected to lie within the above upper and lower limits. Nonlinear least square fitting of the stiffnesses of our most pristine specimens ($\bar{\rho} \geq 37.5\%$) with Eq. (1), assuming a Poisson's ratio $\nu = 0.17$[48] and considering a correction function to account for the structure's face holes (Supplementary Note 4), estimated a constituent material stiffness ($E_s$) of 62 GPa. This agrees remarkably well with the literature utilized upper limit in Fig. 4a, confirming performance of our plate-nanolattices at the HS upper bound. Regarding strength, literature-reported compressive properties[28] of pyrolytic carbon with characteristic dimensions down to 600 nm agree well with our pillar experiments. Literature-reported ultimate tensile strengths of 2 μm-size

specimens of up to 2.75 GPa[29] are also in good agreement, indicating that our pillar-measured yield strengths represent the constituent yield strength of our plate-nanolattices well.

Plate-nanolattices fully exploit the topological advantages of their plate architecture in the higher relative density range. Although all details of the computer models are not fully reflected in our high-$\bar{\rho}$ plate-nanolattices given the limitations of nano-manufacturing, numerical results confirmed that the structural quality of these lattices was sufficient to perform on par with theoretical and numerical predictions, albeit with some experimental scatter. Correspondingly, experiments with polymeric cubic+octet plate-microlattices with near-ideal geometries (Supplementary Note 5) correlated well with the pyrolytic carbon results, but with less scatter. Despite manufacturing imperfections, our pyrolytic carbon data has one of the lowest variabilities among the nanolattices presented in Fig. 6. Outliers in stiffness remain within ±6.9% of the average upper bound predictions.

While plate-lattices clearly outperform beam-lattices in the higher relative density range, our results reveal a tradeoff between performance and manufacturability at lower relative densities. Finite element analyses (Fig. 5) showed that the under-performance of the low-$\bar{\rho}$ plate-nanolattices with respect to the theoretical upper bounds in Fig. 4 is related to small plate thickness-to-size ratios which promote warping during pyrolysis and hence introduce plate curvature via pyrolysis-induced buckling (Fig. 1). Simulations showed that pre-curvature is the key mechanism diminishing stiffness, with a yielding-to-buckling transition at $\bar{\rho} \approx 37.5\%$ also having a strong effect on the strength. As buckling ideally should not affect stiffness measurements, this explains the notably stronger reduction in yield strength compared to stiffness for the low-$\bar{\rho}$ plate-nanolattices. In agreement with the simulations, near-ideal polymer plate-microlattices did not show a transition to worse stiffness scaling at lower relative densities, although the lowest relative density sample displayed reduced strength (Supplementary Note 5). The combination of imperfections and premature buckling of low-$\bar{\rho}$ plate-nanolattices led to convergence of our data with theoretically less efficient pyrolytic carbon beam-nanolattices[6], particularly in terms of strength (Fig. 6). While the manufacturing complexity of nano-architected materials may amplify these effects, low-$\bar{\rho}$ plate-lattices are, regardless of scale, far more difficult to manufacture without significant geometric defects than beam-lattices due to the more extreme aspect ratios of individual features. Moreover, the theoretical performance of plate-lattices improves as the number of plate elements per unit cell is increased (as in the simple cubic + body centered cubic (SC-BCC) design[18]); however, in practice each additional plate element further reduces the lowest manu-facturable relative density, and the simplest topologies like the cubic+octet design may be most practical.

Plate curvature may mitigate brittle failure in plate-lattices. While our high-$\bar{\rho}$, pre-deformation-free specimens failed in a brittle manner, increasing plate curvature at low $\bar{\rho}$ coincided with a more progressive failure behavior, albeit at the cost of reduced mechanical properties (Fig. 3). In previous work, highly curved, thin shell topologies made of brittle constituent materials, such as pyrolytic carbon, have been shown to exhibit progressive failure with high stress plateaus up to 80% strain[45]. This behavior arises from an advantageous crack propagation mechanism displayed by shells with high radius of curvature-to-thickness ratios. In plate-nanolattices, plate curvature induced during pyrolysis also produces such behavior. Interestingly, non-pre-deformed poly-mer cubic+octet plate-microlattices exhibited the same transition from catastrophic to progressive failure (Supplementary Fig. 12) as a result of the introduction of plate curvature by post-yield buckling. Therefore, pre-existent plate curvature is not necessary to induce progressive failure, and elastic buckling may well have

contributed to progressive failure in our pyrolytic carbon struc-tures. The implication is that non-pre-deformed thin-walled pyrolytic carbon plate-nanolattices, if manufacturable, may potentially still exhibit beneficial progressive failure without compromising stiffness and yield strength. Plate-designs with more plate elements per unit cell and thinner features, such as the SC-BCC design, may exploit post-buckling curvature-induced progressive failure most efficiently. At the same time, though, thin wall design is complicated by manufacturing limitations and potentially exacerbates defect-induced buckling strength reduc-tions. Ultimately, both pre-existing and buckling-induced cur-vature are expected to produce a trade-off between compressive strength and deformability.

The combination of an optimal topology at the HS and Suquet upper bounds, and ultra-high strength nanoscale constituent pyrolytic carbon, makes our plate-nanolattices the only cellular material to lie above the theoretical specific strength limit for all bulk materials, as well as outperform all other architected mate-rials in stiffness (Fig. 6). Beam-lattices, such as the octet truss, in practice perform on the order of 25 and 20% of the HS and Suquet upper bounds, respectively; which is in good agreement with the found five-fold and six-fold improvement, respectively, of our plate-nanolattices over TPP-DLW-derived pyrolytic car-bon octet truss and isotropic truss nanolattices[23] in the density range of 0.35–0.79 g/cm³.

In summary, we demonstrated closed-cell plate-nanolattices manufacturable via TPP-DLW and pyrolysis by applying nanometer-size face holes, which we showed to impose only small reductions in mechanical performance, orientation specific hatching strategies and optimization of support structures to minimize shrinkage mismatch with specimens during pyrolysis. This study experimentally demonstrates a cellular material reaching the theoretical strength and stiffness limits of an iso-tropic voided topology. At the same time, the constituent pyr-olytic carbon approaches the theoretical material strength limit, thanks to mechanical size-effects. This combination of both optimal topology and ultra-strong constituent material culmi-nates in the strongest and stiffest existing architected material, with up to 639% average performance improvement over the most efficient beam-nanolattices. While beam-based lattices have dominated structural metamaterials[1] for the past two decades, this study's comprehensive experimental characterization of plate-lattices provides important, scale-independent and process-independent groundwork to establish plate-architecture as a superior design principle. Our results provide understanding of density-dependent performance and deformation behaviors in light of previous theoretical studies and highlight several critical areas of interest for future work, including imperfection sensi-tivity, and design-dependent manufacturing limitations.

Although the TPP-DLW/pyrolysis fabrication route developed in this work is limited to small specimens, plate-lattices with exceptional size-strengthened constituent materials may be manufactured by more scalable techniques (e.g., projection micro-stereolithography[49]) using preceramic resins which exploit size effects into the micrometer range coupled with dramatically reduced shrinkage during pyrolysis[43]. Hence, plate-nanolattices deserve strong attention in the development of scalable high-resolution additive manufacturing processes[50,51], which might in the future make carbon-based nanoarchitected metamaterials as ubiquitous as carbon composites have become in modern engi-neering applications.

## Methods

**Fabrication**. Polymeric plate-nanolattices with $5 \times 5 \times 5$ unit cells were fabricated by two-photon-polymerization direct laser writing (TPP-DLW), with the photo-resist IP-DIP (Nanoscribe GmbH) on silicon substrates, using a Photonic

Professional GT (Nanoscribe GmbH) DLW system. Vertical cubic walls were composed of six 100 nm hatched lines, printed with a laser average power of 12 mW. Horizontal cubic and octet walls were composed of single in-plane voxel lines, printed with laser average powers of 15.25 and 16 mW, respectively. Both octet and vertical cubic walls had 50 nm slicing distances. All walls were printed at a scan speed of 5000 $\mu m\,s^{-1}$. To accommodate otherwise detrimental shrinkage during pyrolysis, structures were printed on multiline coiled springs attached to the top of 340 nm-hatched and 680 nm-sliced support pillars of aspect ratio 0.85. Pillars were printed at a scan speed of 20,000 $\mu m\,s^{-1}$ and an average laser power of 30 mW. Springs were printed at 100 $\mu m\,s^{-1}$ and an average laser power of 7.5 mW. Development was performed in a PGMEA bath for 40 min, whereby introduced plate face holes allowed infiltration of the virtually closed-cell structures with the solvent and removal of enclosed excess raw material. A subsequent 20 min IPA bath was applied to remove residual PGMEA. The polymeric structures were dried in an Autosamdri 931 critical point dryer (Tousimis Research Corp. Inc.) to avoid collapse of the structures by surface tension during evaporation of the solvent. All structures were subsequently pyrolyzed at 900 °C for 1 h in vacuum[6], with a maximum ramp rate of 3 °C min$^{-1}$. During pyrolysis, lattices linearly shrank 78–80%. Separately, individual lattices were printed atop 0.76 mm-diameter tungsten pins and pyrolyzed for imaging by nano-computed tomography. Pillars with 6 $\mu m$-diameters and aspect ratios of 3[52], printed with the same writing parameters of the vertical walls, were fabricated to be representative of the constituent material properties.

**Mechanical characterization**. Relative densities were determined using Solidworks 2018 (Dassault Systèmes SE) and cubic+octet plate-lattice models, including holes, with the SEM measured dimensions of the diagonally-cross-sectioned polymer calibration structures (Supplementary Fig. 3c). The relative density is assumed to not change significantly, given consistent and isotropic shrinkage during pyrolysis[6]. Lattice dimensions were optically measured from high-resolution SEM using a FEI Magellan 400XHR (Thermo Fisher Scientific Inc.). All lattices were mechanically tested under uniaxial compression on an Alemnis Nanoindenter (Alemnis AG) equipped with a 100 $\mu m$ diamond flat punch tip inside a FEI Quanta 3D FEG (Thermo Fisher Scientific Inc.) dual beam (SEM/FIB), using a constant strain rate of 0.001 $s^{-1}$. Front-view in situ SEM videos were used to correct raw load displacement curves for the compliance of the substrate and support pillars via an in-house digital image correlation (DIC) algorithm. An unload-reload cycle was introduced at strains between 4 and 10%, depending on the relative density of the sample, to ensure suspended structures were fully seated on the support pillar surface while avoiding excessive strain-induced sample damage; subsequently, all structures were loaded to failure. Young's moduli were extracted from the maximal slope of linear elastic regimes of the first unloading segment, and yield strengths were determined from the 0.2% yield offset of the engineering stress–strain plots. The compressive strength was measured from the maximal stress before densification. Constituent material properties were measured from in situ pillar compression tests under the same conditions as the plate-nanolattices (Supplementary Fig. 6).

**Microstructural characterization**. Raman spectra were acquired from baked polymeric and pyrolyzed carbon structures (Supplementary Figs. 3a, d, 5) using an inVia (Renishaw plc) confocal Raman microscope with a 50× objective, operated at an excitation wavelength of 532 nm, with a laser intensity of 10% and an exposure time of 1 s over 20 acquisitions. The degree of conversion (DC) was extrapolated from Raman spectra taken of individual polymeric cubic+octet unit cells, pillars printed under the same conditions as the vertical walls, single-voxel horizontal sheets, and octet plate-nanolattice unit cells. DC values were calculated using

$$DC = 1 - \left(\frac{I_{C=C}/I_{C=O}}{I'_{C=C}/I'_{C=O}}\right) \qquad (3)$$

with $I_{C=C}$ and $I_{C=O}$, the integrated intensities of carbon–carbon and carbon–oxygen double bond peaks (1635 cm$^{-1}$ and 1730 cm$^{-1}$, respectively) in the polymerized resin, and $I'_{C=C}$ and $I'_{C=O}$, the integrated intensities of the same peaks in the unpolymerized resin[53] (Supplementary Table 1). The relative degree of graphitization (DG) of pyrolyzed CO plate lattices and pillars was measured via R2 parameter, defined as

$$R2 = \frac{I_{D1}}{I_G + I_{D1} + I_{D2}}, \qquad (4)$$

where $I_G$ (1580 cm$^{-1}$), $I_{D1}$ (1360 cm$^{-1}$), and $I_{D2}$ (1620 cm$^{-1}$) are the integrated intensities of the graphite, first-order defect 1, and first-order defect 2 infrared (IR) bands, respectively, as well as via the half-width-half-maximum (HWHM) of the D1 spectral band[54] (Supplementary Fig. 5; Supplementary Table 2).

**Nano-computed tomography (Nano-CT)**. Samples were scanned in a Xradia Ultra 810 (Carl Zeiss AG) operating with a rotating chromium anode X-ray source with 5.4 keV beam energy, using the phase-contrast imaging mode where the undiffracted X-ray beam is phase-shifted by $3\pi/2$ using a gold phase-ring positioned near the back focal plane of the zone plate. A total of 901 projections were obtained over 180°, with exposure time and detector binning depending on the sample and field of view (Supplementary Table 3). Using the software XMReconstructor (Carl Zeiss AG), the image reconstruction was performed by a filtered back-projection algorithm, the projections were re-aligned for movement compensation using the adaptive mode compensation (AMC) measurements, and the rotation center was adjusted before obtaining the image stacks datasets. The datasets were processed using the software Avizo v. 9.4.0 (Thermo Fisher Scientific Inc.), being first de-noised using its windowed version of the non-local means filter applied in 3D with a search window of 21 voxels, voxel neighborhood of 5 voxels and 1.0 similarity factor, run once for the large field of view (LFOV) 30 and 40% sample datasets and twice for the high resolution (HRES) 60% sample dataset. The segmentation of the image was performed using the watershed algorithm, using marker seeds selected through contrast thresholding, and outside/inside of the carbon cell markers selected via the contrast-based tri-dimensional selection propagation using the magic wand tool, given that phase-contrast gives similar gray levels for the inside and the outside of the carbon cell. Spurious selections due to streaking or thin walls missing from the selection were manually corrected along the sample using the same magic wand tool. The models were subsequently generated from the resulting binary stack using the generate surface module using constrained smoothing 3, which uses an adapted Gaussian filter to smooth voxel edges without losing thin regions in the process, and this surface was re-meshed using the Avizo re-mesh option with high regularity[55], 100% size and fixed contours. The re-meshed surface was then exported as an STL file.

**Finite element analysis**. Models of ideal cubic+octet plate-lattices with and without face holes were constructed in Solidworks 2018 (Dassault Systèmes SE) with SEM-measured wall and hole dimensions of the fabricated polymeric structures. Finite element analysis was performed in ABAQUS (Dassault Systèmes SE) for relative densities of approximately 25, 30, 40, 50, and 60%. Results were determined to be within 1% from a mesh sensitivity study. At mid and low relative densities, computed mechanical properties of shell and solid models do not differ noticeably (Supplementary Fig. 10). Simulations comparing ideal and imperfect, pre-deformed models as well as eigenmode analyses to calculate buckling strengths were carried out in ANSYS 16.2 (Ansys Inc.) with shell models (shell181) with an element size of 2.5% of the unit cell size. Pre-deformations of imperfect models were applied as a percentage of the vertical wall buckling eigenmode of perfect models with zeroed macroscopic displacement. The degree of deformation was adjusted by fitting the computed and experimentally measured average stiffnesses. In all simulations, constituent material properties were representative of nanoscale pyrolytic carbon, with the average yield strength of 2.5 GPa measured from the pyrolytic carbon micro-pillars, the literature reported 62 GPa[27] Young's modulus and a Poisson's ratio of $\nu = 0.17$[56]. An additional set of calculations was performed with $\nu = 0.3$, to facilitate comparison with literature values[21], and to assess the effects of Poisson's ratio on mechanical performance. All models were subject to periodic boundary conditions in the form of uniform macroscopic strains[57,58].

## Data availability
The authors declare that all data supporting the findings of this study are available within the paper and its supplementary information files. The source data underlying Figs. 2a, b, 3a–d, 4a, b, 5a, b, 6a, b and Supplementary Figs. 3a, b, 3d, 5, 6, 7a, b, 8a, b, 9a–d, 10a, b, 12a–f, 13a, b are provided in the Supplementary Data file.

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

## Acknowledgements

This work was financially supported by the Office of Naval Research (program Manager: D. Shifler, Grant No. N00014-17-1-2874). J.B. gratefully acknowledges partial financial support from the Deutsche Forschungsgemeinschaft (DFG), grant BA 5778/1-1. SEM imaging and in-situ mechanical testing was performed at the UC Irvine Materials Research Institute (IMRI), using instrumentation funded in part by the National Science Foundation Center for Chemistry at the Space-Time Limit (CHE-082913). Raman measurements were conducted at the UCI Laser Spectroscopy Lab. The authors are thankful to Dimitry Fishman for useful discussions on Raman spectroscopy. J.M.S.S. acknowledges the Deutsche Forschungsgemeinschaft (DFG) (WE4051/21-1).

## Author contributions

J.B. and L.V. designed the research. C.C. and J.B. developed the manufacturing procedure. C.C. performed micro-Raman spectroscopy. C.C., J.B., and A.G. performed mechanical experiments. C.C. and J.B. analyzed experimental data. C.S.O. and J.M.S.S. performed computed tomography and processed scans. J.B. and J.B.B. carried out finite element analyses. C.C., J.B., and L.V. interpreted results and wrote the manuscript.

## Competing interests

The authors declare no competing interests.
