## [Peer Review File · Nature Communications]

Reviewers' Comments:

Reviewer #1:

Remarks to the Author:

In this manuscript, the authors fabricated the cubic+octet plate-based nanolattices via two photon lithography and pyrolysis and then conducted the in-situ SEM compressive testing on these fabricated pyrolytic carbon plate nanolattices. The results showed that such plate nanolattice exhibited the ultra-high strength and stiffness, approaching theoretical strength and stiffness limits. In particular, the authors overcome multiple manufacturing challenges, including the removal of excess raw materials, orientation specific hatching strategies for wall and optimization of support structures for pyrolysis. The current study provides a route for fabricating the architected materials with superior strength and stiffness. Overall, the current manuscript is considerably comprehensive, well-organized and well-written. Therefore, the reviewer can recommend this manuscript for publication. But the authors have to address the following points to further improve the current manuscript before the paper can be accepted for publication.

(1) Figure 3 shows the compression results of nanolattices with different relative densities, indicating a transition from brittle fracture to progressive deformation with decreasing of relative densities. It is seen in Fig. 3d that for the nanolattice with lower relative density, some plates are curved after pyrolysis. Is the progressive deformation related to the initial curved plates? The authors should address the influence of initial curved plates on the deformation mode and the strength of overall nanolattices.

(2) Figure 3 shows a series of stress-strain curves of nanolattices with different densities. It is seen that the fracture strains of these nanolattices are larger than 20%, exceeding the fracture strains of all previous brittle nanolattices. It is noted that the fracture strain of nanolattice is dependent on the relative density. It is suggested for the authors to address large fracture strain by comparing with other brittle nanolattices, and to add a discussion about the dependence of fracture strain on the relative density.

(3) In the current study, the authors checked the removal of raw materials enclosed within the cells by nano-CT. How are the raw materials after printing removed during fabrication? Is it related to the carbonization during pyrolysis or other manners?

(4) In the third line of third paragraph, page 6, the authors stated "Although our high-rho plate-lattices were nearly defect-free...". What does the defect mean? Does the defect mean the nanovoids and nanocracks (which are the common nanoscale defects in pyrolytic carbon) or other geometrical defects on wall surface?

(5) Recently, several review and perspective papers (Nature Materials, 15, 373, 2016; Advanced Materials, 31, 34, 2018; Small, 1902842, 2019; MRS Bulletin, 44, 750, 2019; MRS Bulletin, 44, 758, 2019; MRS Bulletin, 44, 766, 2019) summarized the design, fabrication and mechanics of various architected materials or mechanical metamaterials (including plate-based lattices), which are very relevant to the current study. The authors are suggested to introduce these review papers to further address the recent advances of various architected materials.

(6) Figure 4 indicates that the stiffness and strength of pyrolytic carbon cubic+octet plate nanolattice reach up to the corresponding theoretical limits. Ref. 18 reported that regardless of any constituent materials and any wall thickness, the cubic+octet plate lattice achieves the Hashin-Shtrikman upper bounds on isotropic elastic stiffness. Did the authors check whether the polymer cubic+octet plate lattice achieves the theoretical stiffness limit?

(7) In the last third line of fourth paragraph, page 4, the authors showed " $b=1.05$ for relative density $\geq 37.5\%$ ", which is not consistent with the data ($b=1.06$) in Table 3.

Reviewer #2:

Remarks to the Author:

This manuscript presents plate-nanolattices exhibiting high stiffness and strength close to theoretical limit. Using two-photon polymerization (TPP) based additive manufacturing method, the authors printed simple cubic+octet (CO) plate-nanolattices with relative density ranging from 25%

to 60%. In order to ensure high quality print with given resolution and printing area, they employed their own orientation-specific hatching strategy. Small holes were incorporated to allow remaining uncured photoresist to be removed after manufacturing. FE analysis showed little effect of these holes on mechanical property. Pyrolysis yielded undistorted carbon nano-lattices, which was confirmed by nano-CT. Stiffness and strength were extracted from stress-strain curves obtained by in-situ mechanical testing. Results showed stiffness and strength reaching theoretical limits. Design, fabrication, and performance test were all carried out with rigor and results are discussed in-depth. However, design for isotropic plate-lattices is not new anymore (e.g. ref. 18), inclusion of holes for manufacturability was reported elsewhere (e.g. ref. 15), and converting additively manufactured polymer to carbon through pyrolysis is quite common technique. The statement that "In this paper, we demonstrate the first plat-nanolattices" is overclaiming as they have been reported (e.g. ref. 15). If the novelty of this work lies on their manufacturing strategy overcoming fabrication challenges for plate-nanolattices, it should be elaborated further and presented more in details, instead of simply presenting hatching diagrams and stating printing parameters were "carefully" selected. Although the authors have overcome several technical difficulties to produce the results presented in this manuscript, what is reported in this manuscript appears to be rather incremental and scientific value/novelty of this work is not considered high. I would not recommend this manuscript for publication.

Reviewer #3:

Remarks to the Author:

This is a well written paper on the influence of both plate geometries as well as size effects on the mechanical properties of architected materials. With the choice of pyrolysed carbon as a building material, the authors were able to achieve record breaking stiffness and strength levels. As such the reviewer considers this publication to be adequate for publication in Nat. Comm., however, only after the following concerns have been addressed.

General comments:

This manuscript claims to lay the ground work to establish plate-architectures as design principles. While what the authors present is novel, other groups (e.g. Mohr et al.) have already introduced plate concepts. The record breaking aspect of this manuscript is the choice of material rather than the introduction of a new design. The authors mention that in the introduction, but the claim in the abstract is tendentially too ambitious.

While the authors nicely show their accomplishment of absolute values in Figure 5, a second new figure with the data normalised by the properties of the monolithic material is required to be able to judge the design benefits.

As the authors are operating at and laudably are able to cope with the limits of spatial resolution of their technique, the design geometries shown in Figure 1a-d are not reflected in the same detail in the actual structures Figure 1 e-l. Please add a paragraph discussing the consequence of the differences between real and ideal geometry. Is there still room for further improvement?

The authors had to introduce holes into the design to allow for reliable fabrication. In Figure 2 the claim and demonstrate that the effect on elastic properties is small and negligible. This, however, is not true for the strengths values, as evidenced by Figure 4b specifically at low relative densities.

Please add a paragraph discussing the issue of brittleness in more detail and how it could be addressed by changes in design or plate thickness. After all the authors would like to explore and benefit from size effects, otherwise the challenge of producing architected materials at such small length scales would be pointless.

Some of the authors have shown in their previous publications on strut geometries that the

polymer lattice could effectively be coated by very thin strong layers. What effect would this approach have in the plate geometry?

At the end of their paper, the authors claim that their approach can be generalised to scalable high-resolution additive manufacturing process. Please substantiate this claim by explaining in detail, how your process at the micron scale can be scaled up or how other processes could benefit from the findings in this manuscript.

Specific comments:

Please page numbers to the manuscript for review.

Figures: labels are too small to be printed on a standard printer, please increase the font.

Figure 5: The authors claim to "penetrate the theoretical limit". A limit can only be reached and not be penetrated. Obviously the lower bound of their theoretical strength limit is too low. Move it up, as it also should not include diamond. Rephrase the caption.

Reviewer #1 (Remarks to the Author):

In this manuscript, the authors fabricated the cubic+octet plate-based nanolattices via two photon lithography and pyrolysis and then conducted the in-situ SEM compressive testing on these fabricated pyrolytic carbon plate nanolattices. The results showed that such plate nanolattice exhibited the ultra-high strength and stiffness, approaching theoretical strength and stiffness limits. In particular, the authors overcome multiple manufacturing challenges, including the removal of excess raw materials, orientation specific hatching strategies for wall and optimization of support structures for pyrolysis. The current study provides a route for fabricating the architected materials with superior strength and stiffness. Overall, the current manuscript is considerably comprehensive, well-organized and well-written. Therefore, the reviewer can recommend this manuscript for publication. But the authors have to address the following points to further improve the current manuscript before the paper can be accepted for publication.

(1.1) Figure 3 shows the compression results of nanolattices with different relative densities, indicating a transition from brittle fracture to progressive deformation with decreasing of relative densities. It is seen in Fig. 3d that for the nanolattice with lower relative density, some plates are curved after pyrolysis. Is the progressive deformation related to the initial curved plates? The authors should address the influence of initial curved plates on the deformation mode and the strength of overall nanolattices.

We thank the reviewer for the suggestion to clarify the influence of manufacturing-induced plate curvature on the mechanical behavior of our plate-nanolattices.

We conducted several additional finite element analyses (Figure 5, lines 168-183, 422-431 & 736-744), which revealed that the observed reduction in the mechanical properties of low relative density plate-nanolattices is indeed related to initial plate curvature. The simulations clearly showed that the key mechanism diminishing stiffness is pre-existent curvature. However, simulations also indicated that a yielding-to-buckling transition occurring at a relative density of ~37.5% had a more severe effect on the strength.

To experimentally investigate the relation between plate curvature and deformation behavior, we fabricated and in-situ mechanically characterized a series of polymeric cubic+octet plate-microlattices with near-ideal, undeformed geometries (Supplementary Information Section 5). Despite ideal geometries, polymer plate-microlattices with relative densities <40% displayed progressive failure, like the pre-curved low-density pyrolytic carbon structures. In situ videos revealed that post-yield (plastic) buckling of sufficiently thin vertical plates introduced plate curvature, leading to progressive failure of the unit cell layers.

As we have recently shown in a separate publication (<https://doi.org/10.1002/sml.201903834>), curvature of unit-cell features plays a central role in progressive failure. Under uniaxial compression, highly curved thin shells exhibit local axial tensile stresses, which promote crack propagation perpendicular to the shell, resulting in sequential failure of small portions of the structures. Conversely, compressed shells with lower radius of curvature-to-thickness ratios exhibit tensile stresses perpendicular to the shell element, which promote catastrophic single crack propagation along the shell through the entire structure. This same mechanism is active in our progressively failing, low-relative density pyrolytic carbon structures. Furthermore, buckling-induced feature curvature also induces progressive failure in near-ideal polymer structures without pre-existent feature curvature. Hence, buckling failure may as well have contributed to the progressive deformation of our pyrolytic carbon structures. In contrast to pre-existent curvature, post-yield buckling-induced curvature of initially straight plates ideally does not affect yield strength and stiffness. Consequently, our polymer plate-microlattices showed no performance knockdown when the failure mode switched from catastrophic to progressive. The important implication is that non-pre-deformed thin-walled pyrolytic carbon plate-nanolattices, if manufacturable, may potentially still exhibit beneficial progressive failure without compromising stiffness and yield strength. This is now discussed in lines 285-303.

In conclusion, our additional experimental and numerical studies allowed us to demonstrate that progressive deformation is strongly related to plate curvature, whether pre-existent or buckling-induced. Additionally, the agreement of our

pyrolytic carbon results with polymer experiments and finite element simulations of ideal geometries reemphasizes that our high-relative density pyrolytic carbon plate-nanolattices were nearly free of geometric defects and correspondingly achieved the theoretical upper bounds of yield strength and stiffness.

(1.2) Figure 3 shows a series of stress-strain curves of nanolattices with different densities. It is seen that the fracture strains of these nanolattices are larger than 20%, exceeding the fracture strains of all previous brittle nanolattices. It is noted that the fracture strain of nanolattice is dependent on the relative density. It is suggested for the authors to address large fracture strain by comparing with other brittle nanolattices, and to add a discussion about the dependence of fracture strain on the relative density.

We thank the reviewer for drawing our attention to this unusually large fracture strain. We realized that we had made a mistake in presenting our data in Figure 3. To accurately determine the mechanical properties of our structures, we corrected the measured strains for the compliance of the test system, substrate and support pillar using digital image correlation (DIC), as described in the methods section. In the submitted manuscript, Figure 3 erroneously showed the non-DIC corrected raw stress-strain curves, which give the impression of unusually large fracture strains. This mistake was corrected in the revised submission.

*The fracture strains of our plate-nanolattices are comparable to those of previously reported pyrolytic carbon beam-nanolattices (Nature Materials **2016**, 15, 4, 438-443; PNAS **2019**, 116, 14, 6665-6672). As reported for many brittle beam-lattices, the failure strain of our plate-nanolattices with relative densities above 37.5% decreases with decreasing $\bar{\rho}$. To understand this behavior, we performed additional strain analysis for $\bar{\rho} \geq 37.5\%$ (Supplementary Figure 7, lines 135-137 & 691-706). Given that stiffness and yield strength both scaled nearly linearly with density, the yield strain was independent of $\bar{\rho}$ and deformation was predominately elastic prior to failure. Newly included computational eigenmode analyses (Figure 5, lines 178-183) suggest the failure strain-relative density dependency is related to post-yield buckling failure. Given that pyrolytic carbon at this scale only exhibits a modest change in stiffness between elastic and plastic deformation, the elastic buckling simulations provide a good approximation for the plastic buckling strength. In good agreement with the measured compressive strength, the computed buckling strength exceeds the yield strength above $\bar{\rho} = 37.5\%$ and quickly increases with $\bar{\rho}$. As a result, the failure strain increases, causing the experimentally observed relationship with $\bar{\rho}$.*

(1.3) In the current study, the authors checked the removal of raw materials enclosed within the cells by nano-CT. How are the raw materials after printing removed during fabrication? Is it related to the carbonization during pyrolysis or other manners?

We apologize this was not made clearer in our original manuscript. In TPP-DLW, structures are printed by selectively polymerizing regions within a droplet of resin. Printed structures are developed in a bath of PGMEA, whereby excess unpolymerized raw material is dissolved. A subsequent bath in IPA is used to remove residual PGMEA. Naturally, closed-cell topologies hinder this process. In this study, we have made the closed-cell cubic+octet structures permeable to solvents by introducing face holes at the center of every plate (see Figure 1). Those holes were just large enough to enable dissolution of the trapped raw material, but small enough to not significantly affect the mechanical properties of the lattices (see Figure 2). Soak times for both PGMEA and IPA were 3-4 times longer than what is typically applied in the case of more accessible open cell structures (e.g., truss lattices). Lastly, structures were dried in a supercritical CO₂ drier to avoid collapsing the polymeric structures by surface tension during evaporation of the solvent. At the end of this process, the material is fully removed from the inside of the cells, and the structure can be pyrolyzed. We have revised the methods fabrication subsection to more clearly explain this raw material removal procedure (lines 346-350).

(1.4) In the third line of third paragraph, page 6, the authors stated “Although our high-rho plate-lattices were nearly defect-free...”. What does the defect mean? Does the defect mean the nanovoids and nanocracks (which are the common nanoscale defects in pyrolytic carbon) or other geometrical defects on wall surface?

We thank the reviewer for this request for clarification. Here we refer to geometric defects and have revised the statements in lines 279-280 to be more precise.

(1.5) Recently, several review and perspective papers (Nature Materials, 15, 373, 2016; Advanced Materials, 31, 34, 2018; Small, 1902842, 2019; MRS Bulletin, 44, 750, 2019; MRS Bulletin, 44, 758, 2019; MRS Bulletin, 44, 766, 2019) summarized the design, fabrication and mechanics of various architected materials or mechanical metamaterials (including plate-based lattices), which are very relevant to the current study. The authors are suggested to introduce these review papers to further address the recent advances of various architected materials.

We thank the reviewer for drawing our attention to the above publications, which we now included in the introduction of state-of-the-art architected materials in lines 38, 49, 52 and 87. Unfortunately, we were not able to find the paper with the citation information of “Advanced Materials, 31, 34, 2018”.

(1.6) Figure 4 indicates that the stiffness and strength of pyrolytic carbon cubic+octet plate nanolattice reach up to the corresponding theoretical limits. Ref. 18 reported that regardless of any constituent materials and any wall thickness, the cubic+octet plate lattice achieves the Hashin–Shtrikman upper bounds on isotropic elastic stiffness. Did the authors check whether the polymer cubic+octet plate lattice achieves the theoretical stiffness limit?

To answer this question (as well as question 1.2 above), we fabricated and in-situ mechanically characterized several polymeric cubic+octet microlattices, which are now described in detail in Supplementary Information Section 5. While the polymer structures reached the Suquet bound, the viscoelastic nature of the polymer prevented us from obtaining a modulus measurement at the Hashin–Shtrikman upper bound (Supplementary Figure 13). The same experimental challenges were noted in Ref. 15. While a prerequisite of the work in Ref. 18 is a linear elastic constituent material, all the highest-resolution additive manufacturing processes (which can make micro- and nano-architected materials) are typically limited to viscoelastic polymers. The additional polymer structure experiments demonstrate that our efforts to realize cubic+octet plate-nanolattices from (linear elastic) pyrolytic carbon were, in addition to the record-breaking aspects, a necessity to demonstrate a material at the Hashin–Shtrikman upper bound via compression experiments. The importance of a linear elastic material for the lattice performance is now pointed out in the main manuscript, lines 207-210.

(1.7) In the last third line of fourth paragraph, page 4, the authors showed “ $b=1.05$ for relative density $\geq 37.5\%$ ”, which is not consistent with the data ($b=1.06$) in Table 4.

We thank the reviewer for identifying this error. The typo has been corrected in the Supplementary Table 4.

Reviewer #2 (Remarks to the Author):

This manuscript presents plate-nanolattices exhibiting high stiffness and strength close to theoretical limit. Using two-photon polymerization (TPP) based additive manufacturing method, the authors printed simple cubic+octet (CO) plate-nanolattices with relative density ranging from 25% to 60%. In order to ensure high quality print with given resolution and printing area, they employed their own orientation-specific hatching strategy. Small holes were incorporated to allow remaining uncured photoresist to be removed after manufacturing. FE analysis showed little effect of these holes on mechanical property. Pyrolysis yielded undistorted carbon nano-lattices, which was confirmed by nano-CT. Stiffness and strength were extracted from stress-strain curves obtained by in-situ mechanical testing. Results showed stiffness and strength reaching theoretical limits. Design, fabrication, and performance test were all carried out with rigor and results are discussed in-depth.

(2.1) However, design for isotropic plate-lattices is not new anymore (e.g. ref. 18), inclusion of holes for manufacturability was reported elsewhere (e.g. ref. 15), and converting additively manufactured polymer to carbon through pyrolysis is quite common technique. The statement that “In this paper, we demonstrate the first plat-nanolattices” is overclaiming as they have been reported (e.g. ref. 15).

The reviewer is certainly correct in stating that the design of our structures has been reported before; however, we respectfully disagree with the derived conclusion regarding the novelty of our work. Whereas the important and relevant results of both Ref. 15 and 18 are purely theoretical, our study represents:

- (i) the first demonstration of a voided material reaching the Hashin–Shtrikman and Suquet upper bounds*
- (ii) the first comprehensive experimental characterization of plate-lattices, providing important, scale- and process-independent groundwork to establish plate-design principles, including density-dependent property and failure behavior, imperfection sensitivity, and design-dependent manufacturing limitations.*
- (iii) the first plate-nanolattices, with only 150-260 nm thick plates, fabricated from pyrolytic carbon by overcoming several significant manufacturing challenges. While certainly “small”, previously fabricated plate-lattices (Ref. 15), which reach neither of the two bounds, are ~22 times larger and solidly micro-scale, with thinnest wall features of 1.5-3.0 μm .*
- (iv) the strongest and stiffest architected materials reported to date, performing up to 639% better than the best nanolattices due to the unprecedented combination of both optimal topology and ultra-strong base material.*

We believe the above key contributions, which are now more clearly summarized in lines 312-326, are impactful enough to warrant publication in Nature Communications.

(2.2) If the novelty of this work lies on their manufacturing strategy overcoming fabrication challenges for plate-nanolattices, it should be elaborated further and presented more in details, instead of simply presenting hatching diagrams and stating printing parameters were “carefully” selected. Although the authors have overcome several technical difficulties to produce the results presented in this manuscript, what is reported in this manuscript appears to be rather incremental and scientific value/novelty of this work is not considered high. I would not recommend this manuscript for publication.

As clarified above, the main contributions of this work are certainly not limited to incremental manufacturing optimizations. That being said, we agree the complexity of our fabrication strategy was vital to the record-breaking nature of our lattices, and we are thankful to the reviewer for pointing out that the manuscript would benefit from a more in-depth description. We have therefore restructured the supplementary information to include Supplementary Section 1, which encompasses all details of the manufacturing strategy employed in this study, including plate orientation-specific Raman spectroscopy measurements of degree of conversion versus writing parameters, wall thickness calibration data, and pyrolysis optimization approaches. We are confident that this additional section will both clearly show the significant progress in nano-manufacturing science and help other researchers duplicate and further improve our processes.

Reviewer #3 (Remarks to the Author):

This is a well written paper on the influence of both plate geometries as well as size effects on the mechanical properties of architected materials. With the choice of pyrolysed carbon as a building material, the authors were able to achieve record breaking stiffness and strength levels. As such the reviewer considers this publication to be adequate for publication in Nat. Comm., however, only after the following concerns have been addressed.

(3.1) This manuscript claims to lay the ground work to establish plate-architectures as design principles. While what the authors present is novel, other groups (e.g. Mohr et al.) have already introduced plate concepts. The record breaking aspect of this manuscript is the choice of material rather than the introduction of a new design. The authors mention that in the introduction, but the claim in the abstract is tendentially too ambitious.

The reviewer is certainly right that plate design principles have been introduced before and that the relevance of our work lies in its experimental achievements. Therefore, we have reworded the abstract as well as the conclusion paragraph in the main text as suggested (lines 23-25 and 321-326). However, being the first comprehensive experimental validation of the plate concept, our study constitutes fundamental groundwork for a design paradigm change from beam- to plate-based architecture, beyond presenting “record-breaking” absolute numbers due to a strong constituent material.

(3.2) While the authors nicely show their accomplishment of absolute values in Figure 5, a second new figure with the data normalised by the properties of the monolithic material is required to be able to judge the design benefits.

As suggested by the reviewer, we have included strength-versus-density and stiffness-versus-density Ashby charts that are normalized by the constituent material properties, to further illustrate the benefits of the cubic+octet plate design over other common architectures (Supplementary Figure 8). Unfortunately, it is often challenging to extract reliable constituent material properties of architected materials from literature, particularly in the case of micro- and nano-architected structures exploiting size effect strengthening, which represent the vast majority of architected materials. Hence, in order to provide meaningful comparisons, we only report data for TPP-DLW-derived pyrolytic carbon architectures. While limited to a single material system, the topologies we present nevertheless include the most relevant beam-architectures, like octet and isotropic trusses, as well as shell spinodal topologies. The cubic+octet topology of our plate-nanolattices can clearly be seen to outperform all other available architectures in both strength and stiffness. This is now pointed out in the main text in lines 200-202.

(3.3) As the authors are operating at and laudably are able to cope with the limits of spatial resolution of their technique, the design geometries shown in Figure 1a-d are not reflected in the same detail in the actual structures Figure 1 e-l. Please add a paragraph discussing the consequence of the differences between real and ideal geometry. Is there still room for further improvement?

The reviewer is correct that the same detail of the computer models is not fully reflected in the pyrolytic carbon plate-nanolattices, particularly in the low relative-density range. To assess the effect of geometric imperfections on the mechanical behavior, we have carried out additional finite element analysis with ideal and imperfect models (Figure 5, lines 168-183, 422-431 & 736-744) and have fabricated and in-situ mechanically characterized a series of polymeric cubic+octet plate-microlattices with near-ideal geometries (Supplementary Information Section 5), complementing our imperfect pyrolytic carbon structures.

Simulations confirmed that the structural quality of our high- $\bar{\rho}$ plate-nanolattices is sufficient to perform on par with theoretical predictions, albeit with some experimental scatter (Figures 5). They also captured the observed reduction in the mechanical properties of low- $\bar{\rho}$ structures related to pyrolysis induced warping, demonstrating that pre-curvature is the key mechanism diminishing the stiffness, with a yielding-to-buckling transition at a relative density of $\sim 37.5\%$ also having a strong effect on the strength. (lines 256-276)

In agreement with the simulations, polymer experiments with near-ideal geometries correlate well with our pyrolytic carbon results, but with reduced scatter. While the viscoelastic nature of the polymer prevented precise measurement of the moduli at the Hashin–Shtrikman upper bound, polymer plate-microlattices did not show a transition in stiffness scaling at low relative density, as the imperfect carbon structures did; this confirmed that the kink in stiffness scaling in carbon plate-nanolattices was due to pre-existing plate curvatures. Polymer specimens did reach the Suquet upper bound, over nearly the entire range of explored densities. Interestingly, despite near-ideal geometries, the deformation behavior of the polymer structures still transitioned from brittle to progressive failure with decreasing $\bar{\rho}$, like their imperfect carbon counterparts. This transition was due to the introduction of plate curvature by post-yield (plastic) buckling of sufficiently thin vertical plates, which enabled progressive failure of a unit cell layer. The important implication is that non-pre-deformed thin-walled pyrolytic carbon plate-nanolattices, if manufacturable, may potentially still exhibit beneficial

progressive failure, without the undesired trade-off of reduced stiffness and yield strength. This is now explained in lines 256-303.

(3.4) The authors had to introduce holes into the design to allow for reliable fabrication. In Figure 2 the claim and demonstrate that the effect on elastic properties is small and negligible. This, however, is not true for the strengths values, as evidenced by Figure 4b specifically at low relative densities.

The reviewer is certainly correct that introducing face holes reduces the strength of the lattices somewhat. However, Mohr et al. (<https://doi.org/10.1002/adma.201803334>) showed in their supplementary information that the effect is not significant for the yield strength. We have included numerical simulations showing that the same applies to the buckling strength. This is now referred to in lines 125-127 and Supplementary Figure 10. As revealed by additionally conducted finite element analysis, the reduction in strength observed at low relative density (where hole effects become less pronounced) is predominantly associated with a yielding-to-buckling transition (Figure 5).

(3.5) Please add a paragraph discussing the issue of brittleness in more detail and how it could be addressed by changes in design or plate thickness. After all the authors would like to explore and benefit from size effects, otherwise the challenge of producing architected materials at such small length scales would be pointless.

The reviewer highlights an interesting aspect of nano-architected materials design. Architecture at small length scales has already been demonstrated as a powerful method to exploit beneficial size effects, but pathways to overcome brittleness are just being investigated. We have revised the discussion section in lines 285-303 to elaborate on plate-curvature-induced progressive, “ductile-like”, deformability.

As we have recently shown in a separate publication (<https://doi.org/10.1002/sml.201903834>), thin-walled, highly curved topologies enable progressive failure with brittle constituent materials due to preferential crack propagation. Under uniaxial compression, highly curved thin shells exhibit local axial tensile stresses which promote crack propagation perpendicular to the shell, resulting in sequential failure of small portions of structures. This same mechanism is active in our progressively failing, low-relative density pyrolytic carbon structures. However, progressive deformation of additionally characterized near-ideal polymer plate-microlattices (Supplementary Section 5) revealed that post-yield buckling-induced curvature was sufficient to cause progressive failure in the absence of pre-existent curvature. In contrast to pre-existent curvature, the polymer plate-microlattices demonstrate post-yield buckling-induced curvature does not reduce the yield strength and stiffness. Ultimately, both pre-existing and buckling-induced curvature are expected to produce a trade-off between compressive strength and deformability.

(3.6) Some of the authors have shown in their previous publications on strut geometries that the polymer lattice could effectively be coated by very thin strong layers. What effect would this approach have in the plate geometry?

While this is certainly an intriguing possibility and despite the presence of the holes included for fabrication, we do not anticipate that atomic layer deposition (ALD) or similar coating processes could coat the structures uniformly, given their virtually closed cell geometry. Even assuming an ideally conformal process, clogging of the small holes would quickly prevent coating of the inside of a structure. Fabrication complications regarding the synthesis of composite lattices are now pointed out in lines 211-214.

Although we are unable to fabricate ceramic-coated carbon plate-nanolattice composites, we expect that the similar mechanical performance of the coating and the already strong pyrolytic carbon would result in only small structural gains, as demonstrated with Al₂O₃-coated pyrolytic carbon beam-nanolattices (Bauer et al., <https://doi.org/10.1038/nmat4561>). In contrast, a strong coating of polymer plate-lattices would certainly facilitate additional strengthening (Bauer et al., <https://doi.org/10.1073/pnas.1315147111>).

(3.7) At the end of their paper, the authors claim that their approach can be generalized to scalable high-resolution additive manufacturing process. Please substantiate this claim by explaining in detail, how your process at the micron scale can be scaled up or how other processes could benefit from the findings in this manuscript.

It was not our intention to claim that the TPP-DLW/pyrolysis process itself was scalable. We apologize for the confusion and have reworded the corresponding section to clarify (lines 327-331). What we wish to convey is that, independent from scale, other processes can directly benefit from several key findings and conclusions of our paper, in particular: (i) performance at the theoretical bounds, (ii) imperfection sensitivity, (iii) density-dependent transition in failure behavior and (iv) design-dependent manufacturing limitations.

While scalability of the “record-breaking” aspects of our structures and constituent material are currently limited, promising potential pathways to increase the scalability of high-resolution additive manufacturing processes in conjunction with preceramic resins are now explained in more detail. For example, projection micro-stereolithography was shown to create centimeter-size parts from architectures with features sizes in the tens of micrometers (<https://www.nature.com/articles/nmat4694>). 3D-printed, polymer resin-derived silicon oxycarbide was demonstrated to offer extreme size-effect strengthening into the range of tens of micrometers (<https://doi.org/10.1016/j.matt.2019.09.009>).

(3.8) Please page numbers to the manuscript for review.

We have included page numbers in the manuscript.

(3.9) Figures: labels are too small to be printed on a standard printer, please increase the font.

This has been addressed in the revised version.

(3.10) Figure 5: The authors claim to "penetrate the theoretical limit". A limit can only be reached and not be penetrated. Obviously the lower bound of their theoretical strength limit is too low. Move it up, as it also should not include diamond. Rephrase the caption.

We apologize for the poor choice of words in the original manuscript. Obviously, the reviewer is correct that penetrating a theoretical limit is an oxymoron. The stronger and weaker upper bounds shown in now Figure 6 assume an arbitrary ideal topology with the best possible scaling of one, for two different constituent materials. The first material (corresponding to the stronger bound) is graphene, the strongest known material at any scale. This represents the theoretical limit for any architected material (at any scale) available today – no structure can do better than this limit, until a constituent material stronger than graphene is identified. The second material (corresponding to the ‘weaker bound’) is diamond, the strongest known bulk material. This represents the theoretical limit for scalable, macro-scale architectures which are manufactured from existing bulk materials. The central motivation for the development of micro- and nano-architected mechanical metamaterials is to create properties which cannot be achieved with large-scale architectures that employ bulk materials. While the weaker bound clearly is not a bound for architected materials in general (as shown by the fact that we violate it with our materials), it is a practical bound for usable materials today and hence is technologically important and useful for evaluating the performance of micro/nano-architected materials (see for example: <https://doi.org/10.1038/nmat4561>, <https://doi.org/10.1073/pnas.1817309116>, <https://doi.org/10.1002/adma.201701850>, <https://doi.org/10.1002/sml.201902842>). We suggest retaining both bounds in now Figure 6 but clarified the meaning of the lower bound in the manuscript (lines 189-192).

Reviewers' Comments:

Reviewer #1:

Remarks to the Author:

After reading the revised manuscript and the responses to three referees, I felt that the authors have addressed nearly all the comments from three referees and that the manuscript has been improved much by adding additional experiments and more discussions and clarification. The current study showed the plate nanolattices with the record-breaking stiffness and strength, which advance the design and fabrication of nanoarchitected materials with excellent mechanical properties. Therefore, I strongly recommended this manuscript for publication. But Refs. 39 and 45 are nearly the same, and the authors should fix it and check the references carefully.

Reviewer #3:

Remarks to the Author:

The authors have fully implemented by suggestions and I strongly recommend publication of this paper in Nature Comm.

Reviewer Comments

Reviewer #1 (Remarks to the Author):

After reading the revised manuscript and the responses to three referees, I felt that the authors have addressed nearly all the comments from three referees and that the manuscript has been improved much by adding additional experiments and more discussions and clarification. The current study showed the plate nanolattices with the record-breaking stiffness and strength, which advance the design and fabrication of nanoarchitected materials with excellent mechanical properties. Therefore, I strongly recommended this manuscript for publication.

1. But Refs. 39 and 45 are nearly the same, and the authors should fix it and check the references carefully.

Ref. 39, which was an erroneous duplicate of ref. 45 has been removed.

Reviewer #3 (Remarks to the Author):

The authors have fully implemented by suggestions and I strongly recommend publication of this paper in Nature Comm.